# Two Directions for Clinical Data Generation with Large Language Models: Data-to-Label and Label-to-Data

**Rumeng Li**[1,2] and **Xun Wang**[3] and **Hong Yu**[1,2,4]
[1]Umass Amherst, Amherst, MA, USA
[2]VA Bedford Healthcare System, Bedford, MA, USA
[3]Microsoft, Redmond, WA, USA
[4]Umass Lowell, Lowell, MA, USA

## Abstract

Large language models (LLMs) can generate natural language texts for various domains and tasks, but their potential for clinical text mining, a domain with scarce, sensitive, and imbalanced medical data, is under-explored. We investigate whether LLMs can augment clinical data for detecting Alzheimer's Disease (AD)-related signs and symptoms from electronic health records (EHRs), a challenging task that requires high expertise. We create a novel pragmatic taxonomy for AD sign and symptom progression based on expert knowledge and generated three datasets: (1) a gold dataset annotated by human experts on longitudinal EHRs of AD patients; (2) a silver dataset created by the data-to-label method, which labels sentences from a public EHR collection with AD-related signs and symptoms; and (3) a bronze dataset created by the label-to-data method which generates sentences with AD-related signs and symptoms based on the label definition. We train a system to detect AD-related signs and symptoms from EHRs. We find that the silver and bronze datasets improves the system performance, outperforming the system using only the gold dataset. This shows that LLMs can generate synthetic clinical data for a complex task by incorporating expert knowledge, and our label-to-data method can produce datasets that are free of sensitive information, while maintaining acceptable quality.

## 1 Introduction

Clinical text holds a large amount of valuable information that is not recorded by the structured data fields in electronic health records (Wang et al., 2018b). Clinical text mining, which aims to extract and analyze information from medical records, such as diagnosis, symptoms, treatments, and outcomes, has various applications, such as clinical decision support, disease surveillance, patient education, and biomedical research (Murdoch and Detsky, 2013). However, clinical text mining faces two major obstacles: the scarcity and sensitivity of medical data. Medical data is often limited in quantity and diversity, due to the high cost and difficulty of data collection and annotation, which require expert knowledge and consent from patients and providers. On the other hand, medical data is highly sensitive and confidential, due to the ethical and legal issues of data privacy and security, which impose strict regulations and restrictions on data collection and usage (Berman, 2002). These obstacles hinder the development and evaluation of clinical text mining methods, especially those based on data-hungry deep learning models.

Recently LLMs have demonstrated impressive performance on many natural language processing (NLP) benchmarks,(Wang et al., 2018a, 2019; Rajpurkar et al., 2016), as well as in medical domain applications (Singhal et al., 2023, 2022; Nori et al., 2023). However, they also face some common problems like hallucination, homogenisation, etc (Azamfirei et al., 2023). Hallucination means that LLMs produce factual errors or inconsistencies in their outputs that do not match the input or the real world. This can damage the reliability and credibility of LLMs, especially for applications in clinical domain that need high accuracy and consistency. In addition, data generated by LLMs tends to be highly homogeneous and fails to capture the diversity and realism of real data, which are essential for many downstream tasks. For example, for clinical text analysis, we need the generated data to cover different types of clinical texts, such as patient histories, diagnoses, or treatment plans. This presents a huge challenge for LLM. The difference between LLM generated data and real data makes people doubt LLMs' practical application value.

In this paper, we investigated whether the outputs of LLMs can be a valuable data source for clinical text mining despite all these aforementioned drawbacks. We focus on Alzheimer's Disease (AD) signs and symptoms detection from electronic

health records (EHRs) notes. Alzheimer's Disease (AD) is a progressive neurodegenerative disorder that affects millions of people worldwide (Scheltens et al., 2021; Schachter and Davis, 2022). It can cause cognitive impairment, behavioral changes, and functional decline. Detecting AD-related signs and symptoms from EHR is a crucial task for early diagnosis, treatment, and care planning (Leifer, 2003). In addition to the scarcity, sensitivity, and imbalance of clinical data, this task is highly challenging due to the high expertise required to interpret the complex and diverse manifestations of AD (Dubois et al., 2021).

We propose a novel pragmatic taxonomy for AD sign and symptom progression based on expert knowledge, which consists of nine categories that capture the cognitive, behavioral, and functional aspects of AD (Bature et al., 2017; Lanctôt et al., 2017). We created three datasets following the taxonomy: (1) a gold dataset annotated by human experts on longitudinal EHRs of AD patients; (2) a silver dataset created by the data-to-label method which labels sentences from a public EHR collection with AD-related signs and symptoms; and (3) a bronze dataset created by the label-to-data method which generates sentences with AD-related signs and symptoms based on the label definition. The "data-to-label" method employs LLMs as annotators and has been widely adopted in many tasks. The "label-to-data", on the other hand, relies on the LLM's generation ability to produce data with labels based on instructions.

We performed experiments of binary classification (whether the sentence is related to AD signs and symptoms or not) and multi-class classification (assign one category from the nine pre-defined categories of AD signs and symptoms to an input sentence), using different data combinations to fine-tune pre-trained language models (PLMs), and we compared their performance on the human annotated gold test set. We observed that the system performances can be significantly improved by the silver and bronze datasets. In particular, combing the gold and bronze dataset, which is generated by the label-to-data method, outperform the model trained only on the gold or gold+silver dataset for some categories. The minority classes with much fewer gold data samples benefit more from the improvement. We noticed slight degradation of performances for a small proportion of categories. But the overall increases in results demonstrates that

LLM can be applied to medical data annotation, and even its hallucinations can be leveraged to create datasets that are free of sensitive information, while preserving acceptable quality.

The contributions of this paper are as follows:

• We create a novel pragmatic taxonomy for AD sign and symptom progression based on expert knowledge, and it has shown to be reliably annotated using information described in EHR notes.

• We investigate whether LLMs can augment clinical data for detecting AD-related signs and symptoms from EHRs, using two different methods: data-to-label and label-to-data.

• We train a system to detect AD-related signs and symptoms from EHRs, using three datasets: gold, silver, and bronze. And evaluate the quality of the synthetic data generated by LLMs using both automatic and human metrics. We show that using the synthetic data improves the system performances, outperforming the system using only the gold dataset.

## 2 Related Work

### 2.1 Large Language Models

Large language models (LLMs) have enabled remarkable advances in many NLP domains because of their excellent results and ability to comprehend natural language. Popular LLMs including GPT-2 (Radford et al., 2019), GPT-3 (Brown et al., 2020), and GPT-4 (OpenAI, 2023), LaMDA (Thoppilan et al., 2022), BLOOM (Scao et al., 2022), LLaMA (Touvron et al., 2023), etc. vary in their model size, data size, training objective, and generation strategy, but they all share the common feature of being able to generate natural language texts across various domains and tasks. They have achieved impressive results on many natural language processing (NLP) benchmarks by leveraging the large-scale and diverse text data from the web.

However, researchers have noticed the drawbacks of LLMs since their debut. Some limitations of LLMs have widely acknowledged and have drawn wide attentions from the research community like hallucination, homogenisation, etc. (Tamkin et al., 2021)

Hallucination is a well-known and widely-studied problem in natural language generation (NLG), which is often defined as "generated content that is nonsensical or unfaithful to the provided source content" (Ji et al., 2023). Hallucination has been observed and analyzed in various NLG tasks,

such as machine translation (MT) (Guerreiro et al., 2023), text summarization (Cao et al., 2021), and dialogue generation (Das et al., 2023). Hallucination can be caused by various factors, such as data noise, model bias, lack of commonsense knowledge, or insufficient supervision. It can be detected and mitigated by various methods, such as data cleaning, model regularization, knowledge injection, or output verification (Ji et al., 2023).

In this paper, we explore a different angle on hallucination, and examine whether the hallucinations of LLMs can be a valuable data source for clinical text processing, rather than a difficulty. We propose that the hallucinations of LLMs can be leveraged to create synthetic or augmented datasets that do not expose sensitive information, but still maintain the linguistic and semantic features of clinical texts, such as vocabulary, syntax, and domain knowledge. Experiments on classification tasks confirmed the validity of the proposal.

Homogenisation is also a potential drawback of using LLMs at a large scale. While it ensures the stability of the text quality, it also reduces the diversity of text. This issue has been noticed and discussed (Marian, 2023), but there is still a lack of research in this direction. In this work, we have observed homogenisation in the "label-to-data" method and conducted experiment which help reveal how it impacts the system performances on the studied task.

## 2.2 Clinical Text Mining and synthetic data generation

Clinical text mining faces two main obstacles: the limited availability and the confidentiality of health data. Various attempts have been done to overcome the lack of and the privacy issues with health data. Public datasets, such as MIMIC (Johnson et al., 2016), i2b2 (Uzuner et al., 2011), or BioASQ (Tsatsaronis et al., 2015) etc, are openly available for research purposes. Synthetic datasets, such as Synthea (Walonoski et al., 2018) and MedGAN (Choi et al., 2017) etc., are constructed based on statistical models, generative models, or rules. They can be used to augment or complement real medical data, without violating the privacy or confidentiality of the patients or providers. Data augmentation or transfer learning techniques are machine learning techniques used to address the data scarcity or imbalance issue by generating or utilizing additional or related data, such as synthetic data, noisy

data, or cross-domain data, to enrich or improve the data representation or diversity (Che et al., 2017; Gligic et al., 2020; Gupta et al., 2018; Xiao et al., 2018; Amin-Nejad et al., 2020; Li et al., 2021).

However, synthetic datasets may not capture the naturalness and realism of human-written medical texts, and may introduce errors or biases that can affect the performance and validity of clinical text mining methods.

LLMs have also been explored for clinical text processing. Research has demonstrated that LLM holds health information (Singhal et al., 2022). Studies has shown that LLMs can generate unstructured data from structured inputs and benefit downstream tasks (Tang et al., 2023). There are also some work that leveraged LLMs for clincal data augmentation (Chintagunta et al., 2021; Guo et al., 2023) In this paper, we propose a novel approach to leverage LLMs, especially its hallucination ability via the label-to-data method as a data source for clinical text processing, which can mitigate the scarcity and sensitivity of medical data.

## 2.3 Alzheimer's disease signs and symptoms detection

Clinical text mining methods have been increasingly applied to detect AD or identify AD signs and symptoms from spontaneous speech or electronic health records, which could be potentially severed as a natural and non-invasive way of assessing cognitive and linguistic functions. (Karlekar et al., 2018) applied neural models to classify and analyze the linguistic characteristics of AD patients using the DementiaBank dataset. (Wang et al., 2021) developed a deep learning model for earlier detection of cognitive decline from clinical notes in EHRs. (Liu and Yuan, 2021) used a novel NLP method based on term frequency-inverse document frequency (TF-IDF) to detect AD from the dialogue contents of the Predictive Challenge of Alzheimer's Disease. (Agbavor and Liang, 2022) used large language models to predict dementia from spontaneous speech, using the DementiaBank and Pitt Corpus datasets. These studies demonstrate the potential of clinical mining methods for assisting diagnosis of AD and analyzing lexical performance in clinical settings.

## 3 Methodology

In this section, we introduce our task of AD signs and symptoms detection, and how we leveraged

LLMs's capabilities for medical data annotation and generation. We also present the three different datasets (gold, silver and bronze) that we created and used for training and evaluating classifiers for AD signs and symptoms detection.

## 3.1 Task overview

Alzheimer's disease (AD) is a neurodegenerative disorder that affects memory, thinking, reasoning, judgment, communication, and behavior. It is the fifth-leading cause of death among Americans age 65 and older (Mucke, 2009). This task aims to identify nine categories of AD signs and symptoms from unstructured clinical notes. The categories are: Cognitive impairment, Notice/concern by others, Require assistance/functional impairment, Physiological changes, Cognitive assessments, Cognitive intervention/therapy, Diagnostic tests, Coping strategy, and Neuropsychiatric symptoms. These categories indicate the stages and severity of AD. Capturing them in unstructured EHRs can help with early diagnosis and intervention, appropriate care and support, disease monitoring and treatment evaluation, and quality of life improvement for people with AD and their caregivers. This is very challenging a task as the AD-related signs and symptoms can vary in form and severity, and it requires a lot of knowledge and experience to capture them from a large amount of text. We ask experts to create the annotation guideline by defining each category of AD-related signs and symptoms and providing examples and instructions for the annotators (See Appendix A for details).

## 3.2 Datasets

As stated above, we created and utilized three different datasets for our experiments: gold, silver and bronze.

### 3.2.1 Gold data (Human annotation)

We expert annotated 5112 longitudinal EHR notes of 76 patients with AD from the U.S. Department of Veterans Affairs Veterans Health Administration (VHA). The use of the data has been approved by the Institutional Review Board at the VHA Bedford Healthcare System, which also approved the waiver of documentation of informed consent. Under physician supervision, two medical professionals annotate the notes for AD signs and symptoms following the annotation guidelines. They selected sentences to be annotated and labelled its categories

as output, and resolved the disagreements by discussion. The inter-annotator agreement was measured by Cohen's kappa as k=0.868, indicating a high level of reliability. This leads to the gold standard dataset with 16,194 sentences with a mean (SD) sentence length of 17.60 (12.69) tokens.

### 3.2.2 Silver data (Data-to-Label)

The silver dataset consisted of 16,069 sentences with a mean (SD) sentence length of 19.60 (15.44) tokens extracted from the MIMIC-III database (Johnson et al., 2016), which is a large collection of de-identified clinical notes from intensive care units. We randomly sampled the sentences from the discharge summaries, and used the LLM model to annotate them with AD-related symptoms. The LLM receives the annotation guidelines and the clinical text as input and produces the sentence to be annotated and its categories as output. The outputs are further checked by the LLM by asking for a reason to explain why the sentence belongs to the assigned category. In this step, the inputs to the LLM are the guidelines and the annotated sentences and the outputs are Boolean values and explanations. This chain-of-thoughts style checking has been proved to improve LLM performances (Wei et al., 2022). Although many LLMs can be used here, we adopt the Llama 65B (Touvron et al., 2023) due to its performances, availability, costs and privacy concerns.

### 3.2.3 Bronze data (Label-to-Data)

We used GPT-4, a state-of-the-art LLM that has been shown to generate coherent and diverse texts across various domains and tasks (OpenAI, 2023). GPT-4 is a transformer-based model with billions of parameters, trained on a large corpus of web texts. We accessed GPT-4 through the Azure OpenAI service [1].

In the generation task, GPT-4 takes only the annotation guidelines as input and produces a piece of note text and outputs the sentence to be annotated and its categories. This is a bronze-level dataset that does not contain any sensitive personal information. It consists of 16,047 sentences with an average sentence length of 16.58 words. Figure 1 shows a snippet of the generated text and annotations. As shown in the example, the model firstly generates a clinical text and then extract sentences of interests for annotation. The generated text is

---

[1]https://azure.microsoft.com/en-us/products/cognitive-services/openai-service/

rich in AD signs and symptoms and contains no Protected Health Information (PHI) or (Personally Identifiable Information) PII. While the text doesn't completely convey the complexity of AD diagnosis, or the follow-up required to arrive at AD diagnosis (e.g. "I immediately drove over and took him to the ER. After a series of tests, including an MRI and a neuropsychological evaluation, he was diagnosed with Alzheimer's disease. " In fact, the diagnosis of AD is a challenging task and it's unlikely to get diagnosed with AD at the ER department), it maintains the linguistic and semantic features of clinical texts, i.e., vocabulary, syntax, and domain knowledge, and represents high quality annotation.

We processed the datasets by tokenizing, lower-casing, and removing duplicate sentences. We split them into train, validation, and test sets (80/10/10 ratio). Table 1 shows the dataset statistics. The gold and silver data have similar average length and standard deviation. The bronze data has a smaller standard deviation and a more balanced categorical distribution than the gold and silver data, which differs from real patient notes. The smaller SD in sentence lengths indicates less diversity in the bronze data. We will experiment with these datasets to see how the LLM's output can help clinical text mining.

### 3.3 Experiments

#### 3.3.1 Classifiers

We use an ensemble method that integrates multiple models and relies on voting to produce the final output. This reduces the variance and bias of individual models and enhances the accuracy and generalization of the prediction, which is crucial for clinical text mining (Mohammed and Kora, 2023).

For the base models, we utilized the power of pre-trained language models (PLMs), which are neural networks that have been trained on large amounts of text data and can capture general linguistic patterns. Three different PLMs, namely BERT (bert-base-uncased) (Devlin et al., 2018) , RoBERTa (roberta-base) (Liu et al., 2019) and ClinicalBERT (Huang et al., 2019) are used in this work. These models have been widely used for clinical text processing and achieved good performances (Vakili et al., 2022; Alsentzer et al., 2019) .

We fine-tune the PLM models on different combinations of the gold, silver, and bronze datasets, as described below. We use cross-entropy loss, Adam optimizer (with a learning rate of 1e-4 and a batch size of 32), and 10 epochs for training. We select the best checkpoints based on the validation accuracy for testing. The outputs are determined by a majority vote strategy.

We use a subset of the gold dataset as the test set. We compare the performances using accuracy, precision, recall, and F1-score.

#### 3.3.2 Data Combinations

**Gold only** We fine-tuned the PLMs only on the training set of the gold data and evaluate the performance on the test set of the gold data.

**Bronze + Gold**, we fine-tuned the PLMs on the bronze data and further fine-tuned them on the training set of the gold data and evaluate the performance on the test set of the gold data.

**Silver + Gold**, we fine-tuned the PLMs on the silver data and further fine-tuned them on the training set of the gold data and evaluate the performance on the test set of the gold data.

**Bronze + Silver + Gold**, we fine-tuned the PLMs on the combination of the bronze data and silver data, and further fine-tuned them on the training set of the gold data and evaluate the performance on the test set of the gold data.

For each data setting, we trained the models as a binary classifier and multi-class classifier. For the binary classification task, we randomly sampled sentences with no AD signs and symptoms from the longitudinal notes of the 76 patients, where our gold data comes from, as negative data. The negative/positive data ratio is 5:1 [2]. The task is to test whether this sentences is AD signs and symptoms relevant or not. For the multi-class classification task, the classifiers need to identify specifically one category that the sentence belongs to among our nine categories of AD signs and symptoms.

## 4 Results and Discussion

### 4.1 Results

Table 2 and Table 3 show the performance of the system on the test set of the gold data, using different combinations of data for fine-tuning PLMs.

The results demonstrate that the system benefits from the silver and bronze data and most increases are significant. For binary classification, the highest performance is obtained by fine-tuning the system on both the bronze+silver data (overall accuracy=0.94, 4.44% ↑), followed by adding

---

[2]See Appendix for negative data generation.

```
|Start of text|
"Some personal history: 3 months ago I received a call from my parents' neighbor that my dad was
    wandering around their property at night. They said he seemed confused and disoriented. I
    immediately drove over and took him to the ER After a series of tests, including an MRI and a
    neuropsychological evaluation, he was diagnosed with Alzheimer's disease. He was prescribed Aricept
    and Namenda, and the doctor recommended that he see an occupational therapist for cognitive
    stimulation. He also suggested some coping strategies, such as using written reminders, keeping a
    daily planner, and doing memory exercises."
|End of text|

|Start of annotation|
[
    {"sentence": "3 months ago I received a call from my parents' neighbor that my dad was wandering
        around their property at night", "class": "Notice/concern by others"},
    {"sentence": "They said he seemed confused and disoriented.", "class": "Cognitive impairment"},
    {"sentence": "After a series of tests, including an MRI and a neuropsychological evaluation, he was
        diagnosed with Alzheimer's disease.", "class": "Cognitive assessment"},
    {"sentence": "He was prescribed Aricept and Namenda, and the doctor recommended that he see an
        occupational therapist for cognitive stimulation.", "class": "Cognitive intervention/therapy"},
    {"sentence": "He also suggested some coping strategies, such as using written reminders, keeping a
        daily planner, and doing memory exercises.", "class": "Coping strategy"}
]
|End of annotation|
```

Figure 1: Illustration of the text and annotation generated by the GPT-4 based on the annotation guideline. (Class names are shown here for better readability.)

| Category | Gold | Silver | Bronze |
|---|---|---|---|
| Cognitive impairment | 6240 | 1378 | 2704 |
| Notice/concern by others | 785 | 366 | 1710 |
| Require assistance | 1864 | 614 | 1205 |
| Physiological changes | 1340 | 5718 | 1769 |
| Cognitive assessments | 2099 | 327 | 2168 |
| Cognitive intervention/therapy | 1097 | 3508 | 2104 |
| Diagnostic tests | 1084 | 2933 | 2021 |
| Coping strategy | 343 | 893 | 1058 |
| Neuropsychiatric symptoms | 1342 | 318 | 1308 |
| Total | 16194 | 16069 | 16047 |
| Avg length +/- SD (tokens) | 17.60 +/- 12.69 | 19.60+/-15.44 | 16.58+/-4.69 |

Table 1: Category Distribution of the Gold/Silver/Bronze Datasets

bronze data only (overall accuracy=0.93 3.33% ↑). The silver data, though derived from MIMIC-III, the real world EHR data, only improves slightly (0.91, 1.11% ↑).

For multi-class classification, the bronze data alone provides the biggest overall performance improvement (7.35%↑). On the contrary the silver data does not improve much (1.47%↑) and even reduces the performance when combined with the bronze data (5.88%↑< 7.35%↑). For sub-categories, the performance gain is large in minority classes including coping strategy (31.82%↑ by adding bronze+silver) and notice/concern by others (21.05%↑ by adding bronze+silver).

## 4.2 Analysis

The performances of machine learning models are largely decided by data quantity and quality. To better understand the results, we conducted a series of analysis.

As Table 1 shows, the gold data has an imbalanced distribution of categories. This poses a challenge for classification tasks. The bronze+silver data, with a more balanced categorical distribution, helps to mitigate this problem. We notice an increase in the performance for Coping strategy (31.82%↑) using bronze+silver data. Performance gains are also observed for other minority classes including NPS, Requires assistance, Cognitive assessment, etc., by adding more training examples.

The amount of data is not the only factor that influences the performance. For the physiological changes category, adding silver data 4 times the size of the gold data makes no difference, while adding a smaller amount of bronze data results in

| | Gold Only | + Bronze | + Silver | + Bronze + Silver |
|---|---|---|---|---|
| Precision (Positive) | 0.73 | **0.88 (20.55%↑)** | 0.73 (0%) | 0.86 (17.81%↑) |
| Recall (Positive) | 0.75 | 0.7 (6.67%↓) | **0.77 (2.67%↑)** | 0.74 (1.33%↓) |
| F-1(Positive) | 0.74 | 0.78 (5.41%↑) | 0.75 (1.35%↑) | **0.8 (8.11%↑)** |
| Overall Accuracy | 0.9 | 0.93 (3.33%↑) | 0.91 (1.11%↑) | **0.94 (4.44%↑)** |

Table 2: Performance (P/R/F-1/Accuracy (change compared to gold only)) of the ensemble system on the gold test set using different data combinations for training (Binary Classification).

| | Gold Only | + Bronze | + Silver | + Bronze + Silver |
|---|---|---|---|---|
| Cognitive impairment | 0.72 | 0.73 (1.4%↑) | 0.72 (0%) | **0.74 (2.78%↑)** |
| Notice/concern by others | 0.38 | 0.4 (5.26%↑) | 0.41 (7.89%↑) | **0.46 (21.05%↑)** |
| Requires assistance | 0.64 | 0.64 (0%) | 0.63 (1.56%↓) | **0.68 (6.25%↑)** |
| Physiological changes | 0.64 | **0.78 (21.88%↑)** | 0.64 (0%) | 0.76 (18.75%↑) |
| Cognitive assessment | 0.69 | 0.75 (8.70%↑) | 0.7 (1.45%↑) | **0.77 (11.59%↑)** |
| Cognitive intervention/therapy | 0.71 | 0.74(4.23%↑)) | 0.72 (1.41%↑) | **0.76 (7.04%↑)** |
| Diagnostic tests | 0.84 | 0.83 (1.19%↓) | **0.87 (3.57%↑)** | 0.82 (2.38%↓) |
| Coping strategy | 0.44 | 0.42 (4.55%↓) | 0.47 (6.82%↑) | **0.58(31.82%↑)** |
| NPS | 0.67 | **0.71 (5.97%↑)** | 0.68 (1.49%↑) | 0.69 (2.99%↑) |
| Overall Accuracy | 0.68 | **0.73 (7.35%↑)** | 0.69 (1.47%↑) | 0.72 (5.88%↑) |

Table 3: Performance (F-1/Accuracy (change compared to gold only)) of the ensemble system on the gold test set using different data combinations for training (Multi-class Classification).

a significant improvement of 21.88% in F-scores. This suggests that the bronze data has a higher quality than the silver data for some categories.

We randomly selected 100 samples from both the silver data and the bronze data and asked our human experts to check the quality of the annotation. The annotation accuracy on bronze data is around 85%, and annotation accuracy on silver data is around 55% indicating the complexity and challenge of real world data. Some examples of LLM's labeling errors are shown in Table 4.

Experts identify at least two types of labelling errors by the LLM in the silver data:

1. Over-inference (example 1&2), the LLM tends to make inference based on the information that is presented, and goes beyond what is supported by the evidence or reasoning.

2. The LLM couldn't handle negation properly (example 3).

In example 1, the LLM infers that the patient is weak so assistance must be required. Similar to example 2, we found that there are some sentences that mentioned son/daughter as nurses also

get labelled as Concerns by others. The LLM infers that specific medical knowledge of children or spouse/children being present at hospital may indicate concern, but this could be wrong.

The mis-classified data impacts the two tasks differently. For binary classification, the system only needs to distinguish sentences with AD-related signs and symptoms from other texts, so the mis-classification is less critical. However, for multi-class classification task, the system needs to correctly assign the categories of the AD-related signs and symptoms, which can be confused by the LLMs outputs. This partially offsets the advantage of increasing the amount of data, especially when using the silver dataset, which has a much lower accuracy than the bronze dataset. We observe that the silver dataset even harms the performance on "Requires Assistance" in multi-class classification task.

On the other hand, when using the bronze dataset, which has a relatively higher quality, we see overall performance improvements for both binary and multi-class classification tasks. We noticed that in the multi-class classification task, the bronze data causes performance degradation on

some categories. The bronze data differs from the gold or silver data, which are real patient notes. This may cause distribution mismatch with the test dataset and lower performance for some categories. Table 1 shows the bronze data has less variation in lengths (4.69 vs 12.69,15.44). This suggests that we need to steer LLMs to produce data that matches the data encountered in practice.

To sum up, different data combinations affect the results (Table 2&3) by varying the training data in amount, quality and distribution. However, the performance generally improves with the addition of the bronze and/or silver data, though further analysis is needed for each category.

## 5 Conclusion and Future Work

In this paper, we examined the possibility of using LLMs for medical data generation, and assessed the effect of LLMs' outputs on clinical text mining. We developed three datasets: a gold dataset annotated by human experts from a medical dataset, which is the most widely used method for clinical data generation, a silver dataset annotated by the LLM from MIMIC (data-to-label) and a bronze dataset generated by the LLM from its hallucinations (label-to-data). We conducted experiments to train classifiers to detect and categorize Alzheimer's disease (AD)-related symptoms from medical records. We discovered that using a combination of gold data plus bronze and/or silver achieved better performances than using gold data only, especially for minority categories, and that the LLM annotations and hallucinations were helpful for augmenting the training data, despite some noise and errors.

Our findings suggest that LLM can be a valuable tool for medical data annotation when used carefully, especially when the data is scarce, sensitive, or costly to obtain and annotate. By using LLM hallucinations, we can create synthetic data that does not contain real patient information, and that can capture some aspects of the clinical language and domain knowledge. However, our approach also has some ethical and practical challenges, such as ensuring the quality, diversity, validity, and reliability of the LLM annotations and hallucinations, protecting the privacy and security of the data and the model, and avoiding the potential harms and biases of the LLM outputs.

For future work, we will investigate other methods and techniques for enhancing and regulating the LLM annotations and hallucinations, such as using prompts, feedback, or adversarial learning. And we would also tackle the ethical and practical issues of using LLM for medical data annotation, by adhering to the best practices and guidelines for responsible and trustworthy AI. We also intend to apply our approach to other clinical text processing tasks, such as relation extraction, entity linking, and clinical note generation.

## 6 Limitations

Despite the promising results, our approach has several limitations that need to be acknowledged and addressed in future work. First, our experiments are based on the experimented LLMs and a single clinical task (AD-related signs and symptoms detection). It is unclear how well our approach can generalize to other LLMs, and other clinical tasks. Different LLMs may have different hallucination patterns and biases, and different clinical tasks may have different annotation criteria and challenges. Therefore, more comprehensive and systematic evaluations are needed to validate the robustness and applicability of our approach.

Second, our approach relies on the quality and quantity of the LLMs annotations and hallucinations, which are not guaranteed to be consistent or accurate. The LLMs produces irrelevant, incorrect, or incomplete annotations or hallucinations, which will introduce noise or confusion to the classifier. Moreover, the LLMs may not cover the full spectrum of the AD-related signs and symptoms, or may generate some rare or novel symptoms that are not in the gold dataset. Therefore, the LLMs' annotations and hallucinations may not fully reflect the true distribution and diversity of the clinical data. To mitigate these issues, we suggest using some quality control mechanisms, such as filtering, sampling, or post-editing, to improve the LLMs' outputs. Fine tuning on high quality gold data can partially address these problems. We also suggest using some data augmentation techniques, such as paraphrasing, synonym substitution, or adversarial perturbation, to enhance the LLMs' outputs.

Third, our approach may raise some ethical and practical concerns regarding the use of LLMs for medical data annotation, especially its hallucinations. Although not observed in this work, there is still a slight possibility that the LLMs may produce some sensitive or personal information that may breach the privacy or consent of the patients or the clinicians. The LLMs may also generate

| No. | Sentence | LLM annotation | Human comments |
|---|---|---|---|
| 1 | [Pt] was profoundly weak, but was no longer tachycardic and had a normal blood pressure. | Requires assistance | Over-inference |
| 2 | Her husband is a pediatric neurologist at [Hospital]. | Notice/concern by others | Over-inference |
| 3 | Neck is supple without lymphadenopathy. | Physiological changes | Miss negation |

Table 4: Examples of the LLM's incorrect annotations from the silver data

some misleading or harmful information that may affect the diagnosis or treatment of the patients or the decision making of the clinicians. Therefore, the LLM outputs should be used with caution and responsibility, and should be verified and validated by human experts before being used for any clinical purposes. We also suggest using some anonymization or encryption techniques to protect the confidentiality and security of the LLM outputs.

# 7 Acknowledgement

This study was supported by the National Institute on Aging of the National Institutes of Health (NIH) under award number R01AG080670. The authors are solely responsible for the content and do not represent the official views of the NIH. We are grateful to Dan Berlowitz, MD from University of Massachusetts Lowell, Brian Silver, MD and Alok Kapoor, MD, from UMass Chan Medical School for their clinical expertise in developing our annotation guidelines of Alzheimer's Disease. We also appreciate the work of our annotators Raelene Goodwin, BS and Heather Keating, PhD and Wen Hu, MS from the Center for Healthcare Organization & Implementation Research, Veterans Affairs Bedford Healthcare System, Bedford, Massachusetts, for developing annotation guidelines and annotating electronic health record notes that were essential for training/evaluating our natural language processing system and assessing the quality of the automatically generated data by LLMs. Finally, we thank our anonymous reviewers and chairs for their constructive comments and feedback that helped us improve our paper.

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

## A    Annotation Guideline

The annotation guideline comprises the main part of the prompts used in this work. It is created by experts and revised based on LLM outputs. The version used in this work is as follows:

```
|Start of annotation schema|

These classes are as follows:

|Class begin|
Class 1:
|Title begin| Cognitive impairment |Title end|
|Definition begin|
(collect broadly, will not be specific to AD).
    Cognitive impairment is when a person has
    trouble remembering, learning,
    concentrating, or making decisions that
    affect their everyday life.
Currently captured by patients subjective
    statements as well as Dr. statements as
    follows:
    Forgetting appointments and dates.
    Forgetting recent conversations and events.
    Having a hard time understanding directions
        or instructions.
    Losing your sense of direction.
    Losing the ability to organize tasks.
    Becoming more impulsive.
    Memory loss.
    Frequently asking the same question or
        repeating the same story over and over
        (perseveration)
    Not recognizing familiar people and places
    Having trouble exercising judgment, such as
        knowing what to do in an emergency
    Difficulty planning and carrying out tasks,
        such as following a recipe or keeping
        track of monthly bills
    meaningless repetition of own words, lack of
        restraint, wandering and getting lost
    lose your train of thought or the thread of
        conversations
    trouble finding your way around familiar
        environments
    problems with speech or language
    feel increasingly overwhelmed by making
        decisions, planning steps to accomplish
        a task or understanding instructions
    mental decline, difficulty thinking and
        understanding, confusion in the evening
        hours, delusion, disorientation, lack
        of orientation, forgetfulness, making
        things up, mental confusion, difficulty
        concentrating, inability to create new
        memories, inability to do simple math,
        or inability to recognize common
        things, poor judgment, impaired
        communication, poor concentration,
        difficulty remembering recent
        conversations, names or events
    forget things more often, forget important
        events such as appointments or social
        engagements, issues with recall,
    changes in abstract reasoning ability
    attention, cognition, speech, orientation,
        judgment
    AD, dementia, MCI: capture diagnoses
```

```
    relevant to this category.
    STM: short term memory loss
|Definition end|
|Class end|

|Class begin|
Class 2:
|Title begin| Notice/concern by others |Title
    end|
|Definition begin|
These concerns are about cognition, mood or
    daily activities, not from nurses or
    doctors or medical care providers, but from
    friends or family or neighbors.
    family complains of something (may be
        related to any class including
        physiology)
    noticed changes in ability, speed
    concern expressed by family/friends
    complaints of pt. easily angered
    some examples:
    Daughter reports that she repeatedly asks
        the same question...had difficulties
        using her smartphone.
    Daughter reports that she has issues with
        banking...some decrease in personal
        hygiene, forgets to take meds, forgets
        where food is in the house, etc.
    Pt. has gone out at 1:30 a.m. without
        telling anyone; they are concerned, but
        pt. always has a response.
    She (daughter) tells me that her mom has
        repeatedly changed the medications in
        the pill boxes that she has arranged
        for her.
|Definition end|
|Class end|

|Class begin|
Class 3:
|Title begin| Requires assistance |Title end|
|Definition begin|
defined as Requires assistance from a person
needs help with or loss of ability with
    ADLs/iADLs, difficulty with self-care,
    trouble managing belongings
    ADLs: dressing, eating, toileting, bathing,
        grooming, mobility
    iADLs: housekeepingrelated activities
        (cleaning, cooking, and laundry) and
        complex activities (telephone use,
        medication intake, use of
        transportation/driving, budget/finance
        management, and shopping)
    some examples:
    The patient will continue to require
        assistance with all complex medical,
        legal and financial decision making.
    She will need 24-hour supervision for her
        safety.
    Direct supervision is required for
        medications using a pillbox.
    Best not to have him use stove.
    If left alone for period of time, will need
        guardian alert or consider camera
        surveillance.
    He is able to make a meal, to dress himself,
        to bathe, to shave, but continues to
        need help with finances.
```

Wife has to remind him about appointments, in particular.
Driving should not be permitted, and he will need assistance with IADLs and decision making.
Veteran does need assistance with all IADLs and most ADLs.
Traveling out of neighborhood, driving, arranging to take buses-limited night driving now
Resides in assisted living facility or nursing home
Writing checks, paying bills, balancing checkbook-minimal (automatic payment) N/A
Playing a game of skill-no hobbies N/A
|Definition end|
|Class end|

|Class begin|
Class 4:
|Title begin| Physiological changes |Title end|
|Definition begin|
senses: vision, hearing, smell loss, SNHL: sensorineural hearing loss, HoH
sleep: Excessive daytime sleepiness, changes in sleep patterns
speech/swallowing (speech difficulties also in "Cognitive Impairment" class)
movement/gait/balance
inability to combine muscle movements: jumbled speech, difficulty speaking, aphasia, dysphasia, difficulty swallowing, dysphagia, difficulty walking, mobility, problems with gait and balance, gait slowing
Brain (and blood vessel-associated) abnormalities
stroke, ischemia, blood vessel occlusion/stenosis, infarct, encephalomalacia, small vessel changes, vascular/microvascular changes (in brain), carotid artery occlusion/disease/atherosclerosis
loss of appetite, loneliness, general discontent, TBI, skull fracture
|Definition end|
|Class end|

|Class begin|
Class 5:
|Title begin| Cognitive assessment |Title end|
|Definition begin|
memory tests, scores irrelevant; mark all present
Blessed Orientation Memory and Concentration (BOMC) test: 0-10 out of 28 is normal to minimally impaired; 11-19 is mild to moderate impairment || VAMC BOMC Scoring: score >10 is consistent with the presence of dementia, score < 7 are considered normal for the elderly
BNT: Boston Naming Test
BVMT-R: Brief Visuospatial Memory Test
CERAD-NAB: Consortium to Establish a Registry for Alzheimer's Disease-Neuropathological Assessment Battery
Clock in a Box

CNS VS: Computerized Neurocognitive Assessment Software Vital Signs
COWAT: Controlled Oral Word Association Test
CVLT: California Verbal Learning Test
DRS: Dementia Rating Scale, Mattis Dementia Rating Scale
D-KEFS: Delis-Kaplan Executive Function System
FAS: a test measuring phonemic word fluency (using words starting with letters F, A, S)
HVLT-R: Hopkins Verbal Learning Test-Revised
HVOT: Hooper Visual Organization Test
Mini Mental State Exam (MMSE; also known as Folstein Test): >=24 and <28 out of 30 (maybe MCI) no CPT code || VAMC MMSE Guidelines: 25-30 normal, 21-24 mild dementia, 13-20 moderate dementia, 0-12 severe dementia || Dr. Peter Morin's scoring: 30 normal, 28-29 MCI, 22-27 mild dementia, 14-21 moderate dementia, 0-13 severe dementia
Montreal Cognitive Assessment (MoCA): >=17 and <26 out of 30 (MCI) free, there is also a Blind MoCA with total score of 21, not 30. || VAMC MoCA Scoring: 26-30 normal, 20-25 suggestive of mild impairment, 15-19 suggestive of moderate impairment, 10-14 suggestive of significant impairment, 0-9 suggestive of severe impairment || Dr. Peter Morin's scoring: 30 normal, 23-26 MCI, 18-22 mild dementia, 10-17 moderate dementia, 0-9 severe dementia
NAB: Neuropsychological Assessment Battery
NBSE: Neurobehavioral status exam (clinical assessment of thinking, reasoning and judgment, e.g., acquired knowledge, attention, language, memory, planning and problem solving, and visual spatial abilities)
NCSE (Cognistat): Neurobehavioral Cognitive Status Exam
NPT/Neuropsych test/neuropsych inventory
PASAT: Paced Auditory Serial Addition Test
Proverb interpretation (test of abstract reasoning; part of MMSE)
RBANS: Repeatable Battery for Assessment of Neuropsychological Status
RCFT: Rey Complex Figure Test (sometimes ROCFT)
RFFT: Ruff Figural Fluency Test
RMT: (Warrington) Recognition Memory Test
Saint Louis University Mental Status Examination (SLUMS): 21-26 out of 30 (MCI) free || VAMC SLUMS Scoring: high school education 27-30 normal, 21-26 mild neurocognitive disorder, 1-20 dementia; less than high school education 25-30 normal, 20-24 mild neurocognitive disorder, 1-19 dementia
SDMT: Symbol Digit Modalities Test a measure of processing speed, concept formation
Serial sevens (part of MMSE)
SILS: Shipley Institute of Living Scale
Spelling a word forward and backward (part of MMSE)
TOMM: Test of Memory Malingering
Trail Making Test
UFOV: Useful Field of View test

```
    VF: Verbal Fluency (test)
    WAIS: Wechsler Adult Intelligence Scale
    WCST: Wisconsin Card Sorting Test
    WTAR: Wechsler Test of Adult Reading

|Definition end|
|Class end|

|Class begin|
Class 6:
|Title begin| Cognitive intervention/therapy
    |Title end|
|Definition begin|
This includes mentions of drugs, doesn't
    require pt to actually start drug or adhere
    to taking drug
    Aricept being taken
    occupational therapy, cognitive linguistic
        therapy, cognitive behavioral therapy
    memory group therapy
    informed pt. of memory group and she had
        possible interest in this
    SmartThink: (regional VA offering) large
        group available to any Veteran who
        would like to improve memory,
        attention, or other cognitive function.
    Dementia-related medications, any
        interventions initiated by provider
        e.g., medications, therapies.
    relevant meds: cholinesterase inhibitors
        (general term), Aducanumab/Aduhelm,
        Memantine/Namenda/Namzaric, Razadyne
        (galantamine), Exelon (rivastigmine),
        Aricept (donepezil)
    Pimavanserin (for
        behavior/agitation/psychosis
        experimental)
    flickering light therapy
    vitamin B12/cyanocobalamin
    vitamin B1/thiamine
    vitamin D/cholecalciferol (in context of
        memory issues only)
|Definition end|
|Class end|

|Class begin|
Class 7:
|Title begin| Diagnostic tests of the head or
    brain that are related to neurocognitive
    symptoms. |Title end|
|Definition begin|
    including CT, EEG, EMG, FDG-PET, MRI, PET,
        PET-CT, MRA, CSF
    MRA=Magnetic resonance angiography
    radiology study (context: header
        neuroimaging)
    imaging (referring to MRI or PET imaging)
    NOT capturing diagnostic test results in
        separate sentences from the test name
    NOT capturing imaging header if specific
        info (MRI) follows
    Include distant MRI (e.g., from childhood);
        concussion/head trauma may be relevant
        to CTE
    genetic testing: APOE4 for sporadic AD,
    mutations in APP, PSEN1 (PS1 protein), PSEN2
        linked to early onset AD
Note MRI in context of spine or joints or EMG
    in context of carpal tunnel syndrome should
    not be considered.
|Definition end|
|Class end|

|Class begin|
Class 8:
|Title begin| Coping strategy |Title end|
|Definition begin|
    repetition and written reminders may be a
        useful tool in therapy
    has been encouraged to keep mentally active
        to slow the rate of cognitive decline
    requires shopping list when going for
        groceries otherwise she will forget
        items
    uses a planner for appointments
    reliant on GPS for driving
    memory exercise
    keep mentally active
    uses medication organizer
|Definition end|
|Class end|

|Class begin|
Class 9:
|Title begin| Neuropsychiatric symptoms |Title
    end|
|Definition begin|
    mood changes: depression, irritability,
        aggression, anxiety, apathy,
        personality changes, behavioral
        changes, agitation
    Feeling increasingly overwhelmed by making
        decisions and plans.
    paranoia, delusions, hallucinations
|Definition end|
|Class end|

|End of annotation schema|
```

## B  Prompts

We used 3 prompts in this work.

1. Prompt 1 is to ask LLM to annotate provided text following the above guidelines.

```
Task: Annotate the text based on the
    provided annotation guideline.

    |Start of text|
    [text here]
    |End of text|

    |Start of annotation guideline|
    [annotation guideline here]
    |End of annotation guideline|
    Format output as a valid json with the
        following structure:
    [
    {
    "sentence":str,\\ The sentence that is
        annotated.
    "class":int \\ The class that the
        sentence belongs to.
    }
    ]
```

2. Prompt 2 is to ask LLM to check the annotation results and explain the reasons for making judgements.

```
Task: Check if the annotations of the text
    based on the provided annotation
    guideline are correct or not and
    explain why.

    |Start of text|
    [text here]
    |End of text|

    |Start of annotation guideline|
    [annotation guideline here]
    |End of annotation guideline|

    |Start of annotation|
    [annotation here]
    |End of annotation|

    Format output as a valid json with the
        following structure:
    [
    {
    "sentence":str,\\ The sentence that is
        annotated
    "class":int, \\ The class that the
        sentence belongs to.
    "decision":bool, \\ Whether the
        annotation is correct or not.
    "reason":str \\ Explain why.
    }
    ]
```

3. Prompt 3 is to ask LLM to generate a note and conduct annotations based on the provided guideline.

```
Task: Generate a clinical note and
    annotate the text based on the
    provided annotation guideline.

    |Start of text|
    [text here]
    |End of text|

    |Start of annotation guideline|
    [annotation guideline here]
    |End of annotation guideline|

    Format annotation output as a valid
        json with the following structure:
    [
    {
    "sentence":str,\\ The sentence that is
        annotated.
    "class":int \\ The class that the
        sentence belongs to.
    }
    ]
```

These prompts are for reference and are slightly modified to adapt to each LLM for format control in practice.

## C   Negative Data Generation

The negative data is sampled from the notes annotated by experts. It consists of data that are not annotated as having any AD-related symptoms. Also sentences that are too short (<5 tokens after removing punctuation and stop words) are removed. Tables/forms/questionnaires are excluded. The ratio of negative:positive data is decided based on statistics from VHA data.

## D   Model Training

The system contains 3 base models. All models are PLMs that are fine tuned on the training data. The models are implemented using Transformers [3]. The training parameters are:

```
epoch=10.
optimizer=Adam.
lr=1e-3.
beats=(0.9, 0.999).
eps=1e-6.
warmup_steps=200.
weight_decay=0.01.
```

---

[3]https://huggingface.co/docs/transformers/index