# OpenReview forum: "Two Directions for Clinical Data Generation with Large Language Models: Data-to-Label and Label-to-Data"
_EMNLP/2023/Conference — EMNLP 2023 Findings_

### Official Review · Reviewer_AkGZ · 2023-08-02

**Soundness:** 2

**Excitement:**

2: Mediocre: This paper makes marginal contributions (vs non-contemporaneous work), so I would rather not see it in the conference.

**Paper Topic And Main Contributions:**

The paper explores the potential use of LLM-generated data to supplement the scarcity, sensitivity, and imbalance issues commonly encountered in medical datasets. However, the paper should provide more rigorous assessment of data quality, address ethical considerations, and include comparisons with existing methods for a well-rounded contribution.

**Reasons To Accept:**

The paper discusses the implications of utilizing LLM-generated data, even with hallucinations, for creating synthetic datasets free of sensitive information.

**Reasons To Reject:**

1. This paper focuses on a specific clinical domain for generating dataset. Yes, we know LLMs can generate data and build a dataset.  It might be novel in clinical scenario, but not novel more broadly.
2. 100 samples for human evaluation is small.

**Reproducibility:**

2: Would be hard pressed to reproduce the results. The contribution depends on data that are simply not available outside the author's institution or consortium; not enough details are provided.

**Reviewer Confidence:**

2: Willing to defend my evaluation, but it is fairly likely that I missed some details, didn't understand some central points, or can't be sure about the novelty of the work.

---

> ### Author Rebuttal · Authors · 2023-08-29
>
> We sincerely thank the reviewer for the valuable comments. We have addressed the comments as follows:
>
> As for assessment of data quality, for our gold data quality, we reported the inter-annotator agreement, which is k=0.868 by Kappa score, indicating a high level of reliability.
>
> The performance boost obtained on the gold test set proved the effectiveness of the generated silver/ bronze dataset. We have conducted human evaluation on a small subset and provided analysis. Due to the high cost, we evaluated another 100 from both silver/bronze datasets and our conclusions still hold. We will include this in the revised version.
>
> For ethical concerns, as stated in the manuscript, this approach aims to address the privacy and security issues in clinical NLP. The MIMIC-III dataset is de-identified and widely used in clinical NLP. The bronze data is purely generated by the model, and can help protect privacy. Moreover, the models we trained in this work are Bert-based, which are small and require less computation resources. They can be easily deployed to assist with clinical AI for wellness and fairness.
>
> As for comparison with existing method, this study presents a novel application, which is a computational approach to detect the early signs of AD. It can be considered as a text classification application. We adopted Bert-based models, which are widely used as the state-of-the-art in clinical text classification tasks [refs: What Clued the AI Doctor In? On the Influence of Data Source and Quality for Transformer-Based Medical Self-Disclosure Detection in EACL2023 ,Bio+Clinical BERT, BERT Base, and CNN Performance Comparison for Predicting Drug-Review Satisfaction in KDD2023 workshop etc.]. We plan to extend our methods to other standardized tasks and available data in the future.
>
> Regarding the novelty, existing work on data augmentation mostly leverage LLMs for tasks like refining and expanding question-answer pairs, to enhance domain-specific QA datasets, (Guo, Zhen, et al), dialogue summarizer datasets (Chintagunta, B et al), NER task (Li, Jianfu, et al.) etc. We presented a novel task, which was to identify nine categories of Alzheimer’s Disease signs and symptoms from unstructured clinical notes. Our work of augmenting training data using synthetic data is one of the steps of creating a high-performance model for identifying the severity of Alzheimer’s disease. In the revision, we will detail the novelty of our task.
>
> The novelty of this work lies in several aspects:
>
> 1.	We proposed a novel pragmatic taxonomy for AD sign and symptom progression. These categories indicate the stages and severity of AD. Capturing them in unstructured EHRs can help with early diagnosis and intervention, appropriate care and support, disease monitoring and treatment evaluation, and quality of life improvement for people with AD and their caregivers. However, existing schema for AD severity identification is challenging in practice, since signs and symptoms are subjective, and in many cases, not present in EHR notes. So we have created a novel taxonomy based on exhaustive literature review and neurologists' professional input. And it has shown to be reliably annotated using information described in EHR notes.
>
> 2.	This task is more challenging than existing LLM tasks in the clinical domains, as it requires classifying data into multiple categories based on complex annotation guidelines (see appendix A). This task demands high levels of in-domain and contextual knowledge. For instance, slow speech problems are usually indicative of cognitive impairment, while other speech problems such as aphasia are related to physiological changes. These distinctions pose a significant challenge to LLMs and machine learning classifiers. We evaluated LLMs on a complex task that needs domain knowledge and language understanding. This work shows how combing expert knowledge (the annotation schema) and LLMs can help with complex tasks.
>
> 3.	This work introduces a novel “label to data” generation scheme, which contrasts with the conventional “data to label” scheme that has been widely used in previous work. We find that this scheme works better under some scenarios, especially in clinical domains where data privacy is highly valued. This suggests a new direction of applying LLMs to clinical tasks. Our proposed approach in creating the silver and bronze data greatly reduces human efforts for annotation.
>
> 4.	The human analysis reveals the problems with LLMs: failure to handle negation and over-inference. These problems, especially the over-inference issue, have not been explicitly pointed out or well addressed in previous work. This work, with the selected examples, demonstrates the limitations of existing LLMs and suggests a possible future research direction.

---

### Official Review · Reviewer_Qqnk · 2023-08-04

**Soundness:** 3

**Excitement:**

3: Ambivalent: It has merits (e.g., it reports state-of-the-art results, the idea is nice), but there are key weaknesses (e.g., it describes incremental work), and it can significantly benefit from another round of revision. However, I won't object to accepting it if my co-reviewers champion it.

**Missing References:**

This relevant work is missing: Li, J., Zhou, Y., Jiang, X., Natarajan, K., Pakhomov, S.V., Liu, H. and Xu, H., 2021. Are synthetic clinical notes useful for real natural language processing tasks: A case study on clinical entity recognition. Journal of the American Medical Informatics Association, 28(10), pp.2193-2201.

**Paper Topic And Main Contributions:**

This paper examined whether combining gold standard dataset with text generated from large language models will improve the performance of information extraction. The authors tested the approach on a unique dataset from VA health and focused on a specific task for Alzheimer's disease signs and symposiums extraction. Their results showed that combining the generated text will improve the performance.

**Questions For The Authors:**

A few specific questions:
1. Line 300. What is the number of sentences that contain positive mentions of signs or symptoms? Is it an imbalanced dataset?
2. Line 331. A period is missing after "16.58".
3. Line 374. Why only BERT models are considered? How about LLMs, such as GPT-4 or LLaMA?


**Reasons To Accept:**

This is a good use case in clinical NLP and empirical evaluation of a hypothesis whether combining gold standard dataset with generated text improve the overall information extraction performance.

**Reasons To Reject:**

The idea of combining synthetic text to enhance NLP performance is not new. Having said that, the application in Alzheimer's disease is new. That being said, this study may be a better fit to informatics community rather than ACL community.

**Reproducibility:**

1: Could not reproduce the results here no matter how hard they tried.

**Reviewer Confidence:**

5: Positive that my evaluation is correct. I read the paper very carefully and I am very familiar with related work.

---

> ### Author Rebuttal · Authors · 2023-08-29
>
> We sincerely thank the reviewer for the valuable comments. We have addressed the comments as follows:
>
> We are grateful for the relevant references suggested by the reviewer; we will update our related work section to include a broader review of related literature. We also appreciate the careful checking and pointing out of typos by the reviewer, which helps us improve our paper.
>
> Regarding the novelty, existing work on data augmentation like the ones you suggested mostly leverage LLMs for tasks like refining and expanding question-answer pairs, to enhance domain-specific QA datasets, (Guo, Zhen, et al), dialogue summarizer datasets (Chintagunta, B et al), NER task (Li, Jianfu, et al.) etc. We presented a novel task, which was to identify nine categories of Alzheimer’s Disease signs and symptoms from unstructured clinical notes. Our work of augmenting training data using synthetic data is one of the steps of creating a high-performance model for identifying the severity of Alzheimer’s disease. In the revision, we will detail the novelty of our task.
>
> The novelty of this work lies in several aspects:
>
> 1.	We proposed a novel pragmatic taxonomy for AD sign and symptom progression. These categories indicate the stages and severity of AD. Capturing them in unstructured EHRs can help with early diagnosis and intervention, appropriate care and support, disease monitoring and treatment evaluation, and quality of life improvement for people with AD and their caregivers. However, existing schema for AD severity identification is challenging in practice, since signs and symptoms are subjective, and in many cases, not present in EHR notes. So we have created a novel taxonomy based on exhaustive literature review and neurologists' professional input. And it has shown to be reliably annotated using information described in EHR notes.
>
> 2.	This task is more challenging than existing LLM tasks in the clinical domains, as it requires classifying data into multiple categories based on complex annotation guidelines (see appendix A). This task demands high levels of in-domain and contextual knowledge. For instance, slow speech problems are usually indicative of cognitive impairment, while other speech problems such as aphasia are related to physiological changes. These distinctions pose a significant challenge to LLMs and machine learning classifiers. We evaluated LLMs on a complex task that needs domain knowledge and language understanding. This work shows how combing expert knowledge (the annotation schema) and LLMs can help with complex tasks.
>
> 3.	This work introduces a novel “label to data” generation scheme, which contrasts with the conventional “data to label” scheme that has been widely used in previous work. We find that this scheme works better under some scenarios, especially in clinical domains where data privacy is highly valued. This suggests a new direction of applying LLMs to clinical tasks. Our proposed approach in creating the silver and bronze data greatly reduces human efforts for annotation.
>
> 4.	The human analysis reveals the problems with LLMs: failure to handle negation and over-inference. These problems, especially the over-inference issue, have not been explicitly pointed out or well addressed in previous work. This work, with the selected examples, demonstrates the limitations of existing LLMs and suggests a possible future research direction.
>
>
> Clinical NLP applications have been an important research topic in the ACL community. The ClinicalNLP workshop and CLPsych workshop have attracted huge interests from the NLP research community. Every year, many influential clinical NLP papers are published in EMNLP and ACL Anthology. A search of "clinical" in ACL Anthology gives about 12,400 results, and "Alzheimer's" returns 955 results. With the applications of NLP techniques in clinical domains, more and more works appear in NLP conferences and we believe this work will add to the attraction of EMNLP and make it a spotlight as AI for healthcare and AI for human well-being.
>
> Here are a few examples of impactful works that appear in ACL Anthology:
>
> "Detecting Linguistic Characteristics of Alzheimer’s Dementia by Interpreting Neural Models" NAACL 2018
>  "Multilingual prediction of Alzheimer’s disease through domain adaptation and concept-based language modelling" NAACL 2019
>  "A Tale of Two Perplexities: Sensitivity of Neural Language Models to Lexical Retrieval Deficits in Dementia of the Alzheimer’s Type" ACL 2020
> "Large language models are few-shot clinical information extractors"  EMNLP2022
>  "ClinicalT5: A Generative Language Model for Clinical Text" ACL2022
>  "MedicalSum: A Guided Clinical Abstractive Summarization Model for Generating Medical Reports from Patient-Doctor Conversations"  EMNLP 2022
>
> About Line 300, We used the MIMIC-III dataset, which contains around 60,000 discharge summary notes, as the source of our silver data. We did not review all of the notes, but randomly sampled some of them until we obtained the desired number of positive instances for our silver dataset. The notes fed to the LLM were highly imbalanced, with sentences containing positive signs and symptoms being relatively rare.
>
> We chose BERT models as our base models for verifying the effectiveness of the generated data, because of the following reasons:
>
> 1) BERT models have been widely adopted in clinical tasks and their effectiveness has been proved in previous works. In contrast, T5 and LLaMA have not yet been widely adopted in the clinical domain.
>
> 2) BERT models have low resource requirements and enable local computation. The use of BERT models makes it possible for on-edge computation, which can facilitate daily personalized health management.
>
> 3) Our gold data is from VHA, which does not allow sending data to any external APIs like GPT-4 for security reasons.
>
> We would further explore leveraging GPT-4 following the privacy guidelines for clinical NLP and meanwhile, we will adopt llama and T5 as our future work.

---

### Official Review · Reviewer_5zTH · 2023-08-05

**Typos Grammar Style And Presentation Improvements:** See point 3 in reasons to reject.
**Soundness:** 3

**Excitement:**

2: Mediocre: This paper makes marginal contributions (vs non-contemporaneous work), so I would rather not see it in the conference.

**Missing References:**

Medically aware GPT-3: https://arxiv.org/pdf/2110.07356.pdf

Dr. Llama: https://arxiv.org/pdf/2305.07804v4.pdf (though in fairness, this might be considered concurrent work)

Exploring Transformer Text Generation for Medical Dataset Augmentation: https://aclanthology.org/2020.lrec-1.578/

**Paper Topic And Main Contributions:**

The paper uses GPT-4 to augment an existing gold dataset with two additional datasets, silver and bronze. Silver is formed through using GPT-4 to label sentences sampled from MIMIC-III, and Bronze is formed through using GPT-4 to create examples that correspond to a specific label. They show that using both silver and bronze datasets help with performance on their classification task in both binary and multi-class classification settings.

Interesting to note: In their setting, they find that the bronze dataset is actually more accurate as judged by humans than the silver dataset, which is surprising and cool.

**Questions For The Authors:**

Can you more thoroughly compare to previous work and explain any methodological novelty of this approach over the previous work (for example for the papers suggested in the missing references section)?

**Reasons To Accept:**

1. The authors perform an interesting comparison of two different types of data augmentation, i.e. either generating the input using the class or generating the class using the input.
2. The results make a compelling case for their method and the combination of different data generation techniques, and the method can help with data imbalance problems, which is especially useful in the clinical domain.
3. The writing is clear and makes a valid argument for the approach.
4. They validate their approach with human evaluations and enumerate two different types of errors.

**Reasons To Reject:**

1. Limited novelty and missing cites: Given the large amount of work on data augmentation using large language models, even in the medical domain (see missing cites), a more thorough related works section is also necessary. Most of the novelty lies mainly in just the application of data generation to this task and the evaluation, which is limited.
2. Limited scope and results: the authors only test this on one dataset and two (related) tasks (binary and multi-class classification). It would be interesting to see if the same procedures (i.e. producing silver labels from inputs and bronze inputs from labels), would induce similar results on a wider variety of tasks. Also, because of the limits in novelty of the approach itself, it might be useful to do a more thorough analysis of the types of errors talked about in section 4.2, maybe through a more thorough manual evaluation, perhaps including chain of thought. Or potentially it might be useful to attempt to remedy some of these mistakes or, for instance the lack of diversity in length of the bronze data. There is also no analysis of the types of errors that occur on the bronze dataset creation, including what the model hallucinations are “beneficial” or potentially “detrimental” to training.
3. Presentation is incomplete and creates confusion: Adding a concept figure seems necessary. Figures and tables could use some major work aesthetically (e.g. less lines in the tables, Figure 1 could be clearer, maybe color code the labels). Additionally, error bars would be welcome in the table, and it is confusing that * represents statistical significance but it looks like this was only calculated for accuracy, so everything else looks like it’s statistically insignificant even though that’s probably not true.

**Reproducibility:**

2: Would be hard pressed to reproduce the results. The contribution depends on data that are simply not available outside the author's institution or consortium; not enough details are provided.

**Reviewer Confidence:**

3: Pretty sure, but there's a chance I missed something. Although I have a good feel for this area in general, I did not carefully check the paper's details, e.g., the math, experimental design, or novelty.

---

> ### Author Rebuttal · Authors · 2023-08-29
>
> We sincerely thank the reviewer for the valuable comments. We have addressed the comments as follows:
>
> We appreciate the relevant references suggested by the reviewer; We will update our related work section to include a broader review of related literature.
>
> Regarding the novelty, existing work on data augmentation like the ones you suggested mostly leverage LLMs for tasks like refining and expanding question-answer pairs, to enhance domain-specific QA datasets, (Guo, Zhen, et al), dialogue summarizer datasets (Chintagunta, B et al), NER task (Li, Jianfu, et al.) etc. We presented a novel task, which was to identify nine categories of Alzheimer’s Disease signs and symptoms from unstructured clinical notes. Our work of augmenting training data using synthetic data is one of the steps of creating a high-performance model for identifying the severity of Alzheimer’s disease. In the revision, we will detail the novelty of our task.
>
> The novelty of this work lies in several aspects:
>
> 1.	We proposed a novel pragmatic taxonomy for AD sign and symptom progression. These categories indicate the stages and severity of AD. Capturing them in unstructured EHRs can help with early diagnosis and intervention, appropriate care and support, disease monitoring and treatment evaluation, and quality of life improvement for people with AD and their caregivers. However, existing schema for AD severity identification is challenging in practice, since signs and symptoms are subjective, and in many cases, not present in EHR notes. So we have created a novel taxonomy based on exhaustive literature review and neurologists' professional input. And it has shown to be reliably annotated using information described in EHR notes.
>
> 2.	This task is more challenging than existing LLM tasks in the clinical domains, as it requires classifying data into multiple categories based on complex annotation guidelines (see appendix A). This task demands high levels of in-domain and contextual knowledge. For instance, slow speech problems are usually indicative of cognitive impairment, while other speech problems such as aphasia are related to physiological changes. These distinctions pose a significant challenge to LLMs and machine learning classifiers. We evaluated LLMs on a complex task that needs domain knowledge and language understanding. This work shows how combing expert knowledge (the annotation schema) and LLMs can help with complex tasks.
>
> 3.	This work introduces a novel “label to data” generation scheme, which contrasts with the conventional “data to label” scheme that has been widely used in previous work. We find that this scheme works better under some scenarios, especially in clinical domains where data privacy is highly valued. This suggests a new direction of applying LLMs to clinical tasks. Our proposed approach in creating the silver and bronze data greatly reduces human efforts for annotation.
>
> 4.	The human analysis reveals the problems with LLMs: failure to handle negation and over-inference. These problems, especially the over-inference issue, have not been explicitly pointed out or well addressed in previous work. This work, with the selected examples, demonstrates the limitations of existing LLMs and suggests a possible future research direction.
>
> We agree with the reviewer and it would be our future work to evaluate the augmentation by synthetic data framework to other tasks for generalizability.
>
> In terms of the evaluation, in our initial submission we evaluated the quality and characteristics of the bronze and silver data using automatic and human analysis.
>
> We reported in the manuscript that
>
> 1.	The bronze data has less diversity as evidenced by the std of length but has a higher accuracy per human evaluation and also more balanced categories. We believe lacking of diversity is detrimental but the higher accuracy and more balanced category distribution are beneficial.
>
> 2.	The sliver data has more diversity (larger std of sentence length) but has a relatively lower accuracy which can be detrimental.
>
> 3.	human analysis showed typical errors made by LLMs in generating data include: failure to handle negation and over-inference.
>
> 4.	We adopted a chain of thought checking for the silver data to increase its annotation quality. As we stated in Section 4.2: "The outputs were further checked by the LLM by asking for a reason to explain why the sentence belongs to the assigned category. In this step, the inputs to the LLM were the guidelines and the annotated sentences and the outputs were Boolean values and explanations. This chain-of-thoughts style checking has been proved to improve LLM performance."
>
> In the revision, we will explain our efforts in improving the diversity and quality of the bronze data.
>
> 1.	We also adopted chain of thought checking for bronze data but did not observe any performance improvements. We suspect it is due to the fact that bronze data are all generated by LLMs and a chain of thought check cannot capture the errors. We will elaborate this in the revised version.
>
> 2.	We have tested different strategies to increase the diversity of bronze data by adding instructions like ”create data of different styles and lengths”. However, as the prompts are very long, LLMs are not sensitive to the diversity/length constraints instructions. This will be our research topic in the future.
>
> We are conducting more thorough human evaluations on the generated data, but it is challenging due to the specialized domain knowledge required.
>
> We will explain these efforts and results in more detail in the revised version. We will also provide more details on how we addressed the LLM errors. We hope that these revisions will address the reviewers' concerns and improve our paper.
>
> We are grateful for your professional suggestions on how to improve our presentation in terms of clarity and aesthetics. We will include them in the final version. We believe these modifications will polish the paper and make it more professional, readable and well-looking.
>
> For simplicity, we only reported the significance tests for the overall accuracies in this table. We will include the significance test results for all metrics and clarify the different tests that we used for different metrics. For precision and recall, we used matched-pair t-tests (sign or Wilcoxon tests are also plausible) that do not require independence assumptions. For F-score, we used a randomization test that is more computationally intensive (ref: More accurate tests for the statistical significance of result differences, Alexander Yeh). And the majority of them were significant. We apologize for not explaining this clearly in the manuscript.

---

### Meta-Review · Area_Chair_B4st · 2023-09-17

**Recommendation:** 3

**Metareview:**

The authors compare two types of data augmentation for the task of detecting Alzheimer's signs and symptoms from clinical notes. They showed that adding in the two augmented datasets (silver and gold) led to an improvement in performance. During the discussion, the authors asserted that the novelty of their work lies mainly in the application area and in the label-to-data direction of generation.

There are comments from numerous reviewers that point to a lack of excitement due to the paper's novelty primarily lying in the application of LLMs to a specific task. While this may get more interest in a medical informatics community (as some reviewers state), it largely does not effect the soundness of the work, and the work seems reasonably connected to NLP. The re-framing that the authors are working on, as well as further discussion of how their findings may apply to other tasks, could make this work more interesting to the NLP community.

---

### Decision · Program_Chairs · 2023-10-07

**Decision:**

Accept-Findings

**Comment:**

The authors compare two types of data augmentation for the task of detecting Alzheimer's signs and symptoms from clinical notes. They showed that adding in the two augmented datasets (silver and gold) led to an improvement in performance. During the discussion, the authors asserted that the novelty of their work lies mainly in the application area and in the label-to-data direction of generation.

There are comments from numerous reviewers that point to a lack of excitement due to the paper's novelty primarily lying in the application of LLMs to a specific task. While this may get more interest in a medical informatics community (as some reviewers state), it largely does not effect the soundness of the work, and the work seems reasonably connected to NLP. The re-framing that the authors are working on, as well as further discussion of how their findings may apply to other tasks, could make this work more interesting to the NLP community.